# The Different Effects of Organic Amines on Synthetic Metal Phosphites/Phosphates

**DOI:** 10.3390/ma13071752

**Published:** 2020-04-09

**Authors:** Xuelei Wang, Zhaojun Dong, Chao Meng, Wei Wang, Hairui Yang, Xizhun Zhuo, Shaobin Yang

**Affiliations:** 1College of Materials Science and Engineering, Liaoning Technical University, Fuxin 123000, China; mikko_mc@163.com (C.M.); wangweilngd@163.com (W.W.); hryanglntu@163.com (H.Y.); 2College of New Energy and Environment, Jilin University, Changchun 130012, China; dongzhaojun@jlu.edu.cn; 3College of Mining, Liaoning Technical University, Fuxin 123000, China; zhuoxizhun@163.com

**Keywords:** metal phosphites/phosphates, crystal structure, helical channel, organic amine, templating effect, solvothermal synthesis

## Abstract

Four metal phosphites/phosphates crystal materials C_8_N_4_H_34_Al_2_P_4_O_18_ (1), C_3_N_2_H_17_GaP_2_O_8_ (2), H_5_In_2_P_3_O_10_ (3), and H_9_In_2_P_3_O_13_ (4) have been solvothermally synthesized by organic amines in the presence of mixed solvents. Structural analyses indicate that compound 1 and 2 show one-dimensional (1D) chain structures; compound 3 and 4 are three-dimensional (3D) inorganic open-framework indium phosphites. Organic amines show different mechanisms in the four compounds. The 2,2′-bipyridine organic amine acts as a template source and it breaks down small molecules, which enter into the structure of compound 1. For compound 2, 1,2-propanediamine has a role as protonated template and it forms a hydrogen bond with the inorganic skeleton structure. As for compound 3 and 4 without the organic template, the benzylamine and 2,2′-bipyridine mainly serve as structure-directing agent. Especially, compound 3 has an odd seven-ring channel, and compound 4 contains 3D intersecting six-ring, eight-ring, and 10-ring channels. X-ray diffraction (XRD), scanning electron microscopy (SEM), CHN, inductive coupled plasma (ICP), Infrared (IR), and thermal gravimetric (TG) analyze the four compounds.

## 1. Introduction

Inorganic open-framework metal phosphates have been widely studied due to their channel structure and potential applications in industrial catalysis, gas separations, and so forth [1,2,3,4]. For example, SAPO-34 catalysts provide an alternative platform for methanol conversion to olefins (MTO), which produces basic chemicals from nonparallel resources, such as natural gas and coal [5]. Since the aluminophosphate zeolite was first reported in 1982 [6], various metal phosphate materials have been found in related literature [7,8,9]. Presently, a lot of researches on pyramidal phosphite have been made for replacing traditional phosphate tetrahedron [10,11,12]. Novel structures with an interrupted framework can be easily formed due to the P-H bond of phosphite [13]. Particularly, some novel structures and performances of metal phosphites can be easily generated, such as ZnHPO-CJ1 with extra-large 24-ring channels [14], JIS-3 containing magnetic anisotropy of the antiferromagnetism [15], NTHU-13 with tunable channel sizes that range from 24-ring to 72-ring channels [16] and QDU-5 involving room temperature phosphorescence [17]. The results show that the account of transition metal phosphites is major in the reported literature on phosphites. However, the main group metal phosphite has been less reported [18], exploring an effective route to prepare main group metal phosphite materials is of particular importance.

Generally, organic amines are required to synthesize open framework metal phosphites/phosphates materials [19]. Various amines play different functions in the preparation process of abundant open framework structures. For instance, the methylviologen played as protonated templates in JU98 and JU104 [20,21], the 1,3,5-tri (1-imidazolyl) phenyl organic amine acted as ligands in NCU-2 [22], and the 4-dimethylaminopyridine personated as structure-directing agent in PKU-25Al [23]. Organic amines can play multiple roles in the formation of crystal materials [24,25]. In addition, some organic amines have not been integrated into the structure, but the target product cannot be synthesized without the addition of organic amines, such as the structure of InHPO-CJ13 [26]. Special porous channels and chemical properties are constituted by the presence of organic amines [27]. Exploring the role of organic amines is meaningful in the synthetic field of open-framework metal phosphites/phosphates. 

We systematically designed a series of experimental ratios to explore the synthesis of Al, Ga, and In open-framework phosphites/phosphates materials in order to study the main group of metal phosphite materials. Herein, Four main group metal phosphites/phosphates crystal materials C_8_N_4_H_34_Al_2_P_4_O_18_ (1), C_3_N_2_H_17_GaP_2_O_8_ (2), H_5_In_2_P_3_O_10_ (3), and H_11_In_2_P_3_O_13_ (4) with different dimensional features were solvothermally synthesized by organic amines. In addition to the synthesis of four new materials, different from the previous synthesis of main group metal phosphite materials [28], we also analyzed the mechanism of organic amine in the four materials. Interestingly, these organic amines show diverse manifestations in the synthetic process.

## 2. Materials and Methods 

### 2.1. Materials and Instruments

The reagents were purchased and not further purified. These mainly contained phosphorous acid (H_3_PO_3_, 99%, Aladdin Chemistry Co., Ltd., Shanghai, China), aluminium isopropoxide (C_9_H_21_AlO_3_, 98%, Aladdin Chemistry Co., Ltd., Shanghai, China), gallium nitrate (Ga(NO_3_)_3_·6H_2_O, 98%, Sinopharm Chemical Reagent Co., Ltd., Shanghai, China), indium chloride (InCl_3_·4H_2_O, 98%, Sinopharm Chemical Reagent Co., Ltd., Shanghai, China), 2,2′-bipyridine (C_10_H_8_N_2_, 99%, 1.1668 g/ml, Sinopharm Chemical Reagent Co., Ltd., Shanghai, China), 1,2-propanediamine (C_3_H_10_N_2_, 99%, 0.86 g/ml, Sinopharm Chemical Reagent Co., Ltd., Shanghai, China), benzylamine (C_7_H_9_N, 99%, 0.98 g/ml, Sinopharm Chemical Reagent Co., Ltd., Shanghai, China), ethanol (C_2_H_6_O, 99.5%, 0.79 g/ml, Liaoning Quan Rui Reagent Co., Ltd., Shenyang, China), 1,2-propanediol (C_3_H_8_O_2_, 99%, 1.04 g/ml, Liaoning Quan Rui Reagent Co., Ltd., Shenyang, China), isobutanol (C_4_H_10_O, 99%, 0.81 g/ml, Liaoning Quan Rui Reagent Co., Ltd., Shenyang, China), 1-pentanol (C_5_H_12_O, 99%, 0.82 g/ml, Liaoning Quan Rui Reagent Co., Ltd., Shenyang, China), and demonized water (18.2 MΩ·cm, N50-DI Deionizer, Dongguanshi Naibaichuan Water Treatment Equipment Co., Ltd., Dongguan, China). 

The powder X-ray diffraction (XRD) data were measured by using Rigaku 2500 X-ray diffractometer (λ = 1.5418 Å) device (Rigaku, Tokyo, Japan). Thermal gravimetric (TG) analyses of samples were performed on a device of Perkin-Elmer TGA7 with a heating rate of 10 °C/min. (Perkin-Elmer, Waltham, USA). Infrared spectrum (IR) data were detected on Nicolet Impact 410 FTIR spectrometer in a wavenumber range of 4000–400 cm^−1^ (Nicolet, New York, USA). Al, Ga, In and P elements were analyzed using the Perkin-Elmer Optima 3300 DV inductive coupled plasma (ICP) spectrometer (Perkin-Elmer, Waltham, USA). C, N, and H elements were conducted on a Perkin-Elmer 2400 elemental (CHN) analyzer (Perkin-Elmer, Waltham, USA). The surface morphology of samples was characterized by using JEOL JSM-7500F Scanning electron microscopy (SEM) device (JEOL, Tokyo, Japan). The single-crystal XRD data were tested by using Bruker ApexII CCD diffractometer (Bruker, Karlsruhe, Germany). 

### 2.2. Synthesis and Initial Characterization

In a typical reaction to synthesize **1**, a mixture of aluminium isopropoxide (0.10 g, 0.5 mmol), 2,2′-bipyridine (0.16 g, 1.0 mmol), phosphite (0.328 g, 4.0 mmol), ethanol (1 mL), and demonized water (8 mL) was put into a reaction kettle at 160 °C for 7 d. The colorless cube like products were solvothermally synthesized, being rinsed twice with demonized water and then dried in the air (Figure 1a). The results of simulation and experimental XRD show that the product has a pure phase (Figure 2a). The ICP and CHN analyses of C_8_N_4_H_34_Al_2_P_4_O_18_ (**1**) (651.96) gave P: 19.05%, Al: 8.22%, C: 14.67%, H: 5.16%, and N: 8.64%. (Calc. P: 19.00%, Al: 8.31%, C: 14.74%, H: 5.22% and N: 8.60%). The IR spectrum of 1 exhibited typical curve (Figure 3a). The spectrum range of 2860–3230 cm^−1^ was owing to stretching vibrations of N-H bonds and C-H bonds, while the peaks at 1205, 1006, 576, and 545 cm^−1^ were related to the vibration of the P-O bonds [29]. The absorption peaks at 1630, 3399, and 3548 cm^−1^ were due to the bending and stretching vibrations of O–H bonds.

To synthesize 2, a typical synthetic process of Ga(NO_3_)_3_·6H_2_O (0.12 g), H_3_PO_3_ (0.25 g), 1,2-propanediamine (0.25 mL), 1,2-propanediol (6 mL), and demonized water (3 mL) under stirring to form a mixed solution. The final mixture was put into the stainless steel reactor and kept at 160 °C for 5 d. The colorless parallelepiped block products were rinsed and collected (Figure 1b). The characteristic peaks of simulated and experimental XRD analysis were corresponding (Figure 2b). Experimental test analysis found (%) C_3_N_2_H_17_GaP_2_O_8_ (**2**) (340.83): P, 14.35; Ga, 29.78; C, 10.62; H, 4.87; N, 8.15. Calc. (%): P, 14.24; Ga, 30.08; C, 10.56; H, 4.99; N, 8.22. The IR spectrum of **2** (Figure 3b) gave the absorption peaks at 3220 cm^−1^ and 3130 cm^−1^, which could be classified as a stretching vibration of O-H and N-H bonds. The absorption peak at 2960 cm^−1^ was attributed to the stretching vibration of C-H bonds, while a sharp absorption peak at 2370 cm^−1^ proved the presence of the P–H group [30]. The sharp absorption peaks at 1620 cm^−1^ and 1560 cm^−1^ could be resulted from the bending vibration of C-H and N-H bonds. The peaks at 1120 cm^−1^ and 540 cm^−1^ were brought about the stretching and bending vibration of the P-O bonds.

Compound 3 was solvothermally prepared from mixture of InCl_3_·4H_2_O, H_3_PO_3_, benzylamine, isobutanol, and demonized water with the mole ratio of 1.0 : 16.0 : 4.0 : 200 : 1000. The reaction mixture was enclosed into the autoclave and heated at 160 °C for 7 d. Subsequently, the hexahedral block products were rinsed and collected (Figure 1c). The positions of characteristic peak for simulated and experimental XRD are consistent (Figure 2c). Elemental analysis found (%) H_5_In_2_P_3_O_10_ (**3**) (487.59): P, 19.05; In, 47.10; H, 1.03. Calc. (%): P, 19.13; In, 47.12; H, 1.01. The IR spectrum (Figure 3c) exhibited typical changes for **3**. The interrupted bands at 1080, 590, and 495 cm^−1^ were owing to the vibration of P-O bonds. The sharp band of 2370 cm^−1^ confirmed the existence of P–H groups. The absorption peaks at 3330 cm^−1^, 3250 cm^−1^, and 3160 cm^−1^ were related to the stretching vibration of O-H bonds.

Compound 4 was prepared in the system of InCl_3_·4H_2_O-2,2′-bipyridine-H_3_PO_3_-1-pentanol-H_2_O. Typically, InCl_3_·4H_2_O (0.11 g), 2,2′-bipyridine (0.16 g), H_3_PO_3_ (0.33 g), and 1-pentanol (3.0 mL) were dissolved into demonized water (5.0 mL). The mixtures were put into the reactor at 150 °C for 5 d. The hexahedral block products were rinsed and collected to analyze data after drying in air (Figure 1d). The positions of characteristic peak for the simulated and experimental XRD are consistent (Figure 2d). Elemental analysis found (%) H_11_In_2_P_3_O_13_ (4) (541.62): P, 17.25; In, 42.46; H, 1.76. Calc. (%): P, 17.16; In, 42.39; H, 1.67. The IR spectrum of compound 4 (Figure 3d) exhibited a typical structural functional group changes. The discontinuous bands at 1110, 570, and 510 cm^−1^ were owing to the vibration of P-O bonds. The presence of P–H groups with stretching vibration was proven by the peak at 2420 cm^−1^. The absorption bands at 3245 cm^−1^, 3150 cm^−1^, and 2980 cm^−1^ were classified as the stretching vibration of O-H bond.

### 2.3. Determination of Crystal Structure

Four massive single crystals were used to test single-crystal X-ray diffraction analyses. The SAINT program treated data processing [31]. Their crystal structures were repaired by using refined via full matrix least squares of SHELXTL-97 and direct methods [32]. For compound 1 and 2, the atoms of inorganic skeleton and organic matter were easily located. However, organic amine of compound 1 was decomposed, and confirmed by analyses of CHN, IR, and TG. While for compound 3 and 4, the structure contained no organic template. The locations of inorganic skeleton atoms could be easily determined, and other homologous atoms were decided by the Fourier maps. Table 1 shows the experimental data of four compounds for the structural decision. Appendix A provides the data of bond distances and bond angles data. The CCDC numbers (1985961–1985964) can be obtained from www.ccdc.cam.ac.uk/data_request/cif.

## 3. Results and Discussion

### 3.1. Structure and Characterization

C_8_N_4_H_34_Al_2_P_4_O_18_, 1. The asymmetric unit includes one phosphorus atom, one aluminum atom, two nitrogen atoms, four carbon atoms, seven oxygen atoms, and one free water molecule (Figure 4a). The space between the P atom and O atom is in the region 1.504(3)–1.540(3) Å, while the bond angles of O-P-O vary in the region of 106.89(17)–113.16(18)°. The Al-O bonds are located in the range from 1.717(3) Å to 1.722(3) Å. The O-Al-O bond angle varies in the region of 106.07(16)–113.5(2)° (Appendix A), which is related to the four coordinated Al atom. These structural data are compatible with publicly reported in other aluminum phosphates [33]. P atom and Al atom are both paired by four oxygen atoms. The alternating connection of the PO_4_ tetrahedras and AlO_4_ tetrahedras leads to generate a one-dimensional (1D) chain with vertex-sharing connected four-ring along z axis (Figure 5a). Protonized organic amines and lattice water molecules balance the charges of anion skeleton. We can note that the organic amines and 1D chains are arranged in the ABAB alternate dislocation form along [010] direction (Figure 5b). A similar 1D chain has been recorded in the phosphate literature [34]. It is worth noting that the original organic template was broken down and it entered the structure in the form of another new type, which is a relatively rare phenomenon. This phenomenon has not been reported in the synthesis of one-dimensional aluminum phosphate.

C_3_N_2_H_17_GaP_2_O_8_, 2. The asymmetric unit contains one GaO_6_ octahedron, two HPO_3_ pseudo pyramids, one diprotonized 1,2-propanediamine molecule, and one free water molecule (Figure 4b). What is interesting is that there is a hydroxyl group to coordinate the Ga of octahedron. The Ga–O bond distance is located in the region 1.941(4)–1.998(4) Å and O-Ga-O bond angle varies from 86.92(19)° to 179.62(19)°. For the HPO_3_ pseudo pyramids, the P-H bond lengths are 1.3333 Å and 1.3721 Å, and the P-O bond distance is situated in the ranges from 1.497(5)–1.524(5) Å. The O-H bond length is 0.9863 Å, while other structural data were provided by Appendix A. In terms of dimension, 2 has a particular 1D chain structure, which contains Ga-O-Ga chain (Figure 6a). The hydroxyl group is directly linked between the GaO_6_ octahedron. The hydrogen bonds are formed between diprotonized n-heptylamine and framework chain (Figure 6b), a pseudo-2D layer structure is constructed from hydrogen bond connections. A similar 1D chain aluminum phosphite has been reported in known literature [35]. However, no similar structure has been reported in gallium phosphite compounds. There is no correlative literature on the synthesis of one-dimensional chain gallium phosphite using 1,2-propanediamine molecule.

H_5_In_2_P_3_O_10_, **3**. It contains three crystallographically distinct P atoms, two crystallographically unique In atoms, nine oxygen atoms, and one coordinate water molecule (Figure 4c). All of the P atoms are coordinated to three O atoms and a terminal H atom. The P-O bond varies from 1.505(3) Å to 1.558(3) Å and O-P-O bond angles are in the range of 103.7–114.90(18)°. The P-H bond lengths are 1.3205 Å, 1.2239 Å, and 1.3038 Å. Each of the In atoms is octahedrally coordinated to six oxygen atoms. The In-O distance varies from 2.072(3) Å to 2.237(3) Å, while the O-In-O bond angle changes from 84.64(11)° to 170.72(13)° (Appendix A). **3** is a 3D inorganic open-framework indium phosphite compound. It shows a pillared-layer structure along [001] direction (Figure 7a). The layers are connected to each other by bridging oxygen. The layer structure consists of dense four-ring in (101) plane, as indicated in Figure 7b. It can also be viewed as a 1D chain ABAB stack, which contains a special binuclear In_2_O_10_ cluster (Figure 7c). We can find that In_2_O_10_ cluster is joined by edge-sharing, and it forms a four-ring chain structure by dislocation connection with HPO_3_ pseudo pyramids. Rarely, compound 3 has an odd seven-ring channel in the (100) direction and it contains two kinds of channels with different positions (Figure 8). A similar three-dimensional (3D) open-framework compound has been reported in indium phosphite literature [36]. The difference is that the used organic templating agent is different, and the structural analysis method is different, especially the seven-ring channel.

H_9_In_2_P_3_O_13_, 4. As indicated in Figure 4d, it contains three crystallographically unique phosphorus atoms, two unique indium atoms, nine crystallographically unique oxygen atoms, one free water molecule, and two coordinate water molecules. All the phosphorus atoms exist in the form of HPO_3_ pseudo pyramids, while indium atoms exist in the form of InO_6_ octahedron. The distance of P-O bonds is in the range of 1.504(3)–1.532(4) Å, In-O bond distances vary from 2.084(3) Å to 2.203(3) Å, and the P-H bond lengths are 1.3711 Å, 1.3383 Å, and 1.3340 Å. Refer to Appendix A for the detailed structure information of 4. Compound 4 is 3D open-framework structure with 3D intersecting six-ring, eight-ring, and 10-ring channels along (001), (100), and (010) directions (Figure 9). When following the [001] direction, it can be considered as a densely packed ABAB layer structure (Figure 10a). The relationship between AB layers is shown in Figure 11. The layers have 12-ring windows, which are stacked by chains and HPO_3_ pillars (Figure 10b). It is noteworthy that the chain is a unique left-handed helical chain (Figure 10c). The isostructural inorganic structure has been appeared in related literature report [37]. The difference is the used organic amine and the analysis of the 12-ring window.

### 3.2. Thermogravimetric Analysis

Figure 12 presents the derivative of losses for compound 1–4. The TG curve of 1 exhibited a total weight loss of 30.63%, which was due to the miss of lattice water molecules (observed: 4.68%; calculated: 5.52%) and organic amine molecules (observed: 25.95%; calculated: 27.04%) (Figure 12a). The lower weight loss is due to the remaining carbon in the solid residue, which cannot burn out (black in color) [38]. TG for 2 showed three stages of weight loss with a total of 33.17% between 100 and 650 °C (Figure 12b), which was related to the lattice water molecules (observed: 5.26%; calculated: 5.29%), 1,2-propanediamine molecules (observed: 24.37%; calculated: 21.75%), and hydroxyl groups (observed: 3.53%; calculated: 4.99%). The hydroxyl groups were released as water molecules. In addition, sectional hydroxyl groups were released in the second weightlessness temperature region. For compound 3, the observed decrease of 3.64% between 100 and 400 °C corresponded well to the coordinated water (calculated: 3.69%) (Figure 12c). The TG curve of 4 showed a total weight decrease of 13.46%, which was related to the lattice water molecules (observed: 3.99%; calculated: 3.34%) and coordinated water molecules (observed: 9.35%; calculated: 10.02%) (Figure 12d). The XRD analysis indicates that the structure of 1–4 became amorphous after heat treatment at 400 °C for 2 h. IR analysis could indicate that the water and propanediamine of 1–4 had been removed after heat treatment at the corresponding temperature. 

### 3.3. Roles of Organic Amines

These four different compounds were synthesized by three organic amines. These organic amines, as alkaline substances, will play a role in regulating the pH value. In addition, they have different functions in different forms. For compound 1, the 2,2′-bipyridine organic template breaks down 2,3-diaminobutane small molecule, which enters into the structure. It is part of the molecular 2, 2-bipyridine structure, as seen from the structure. Accordingly, 2,2′-bipyridine acts as a template source. As for compound 2, 1,2-propanediamine organic amine goes into the structure, and retained intact. It has a role as protonated template to balance the anion charge of the skeleton. On the other side, the N-H bonds of the 1,2-propanediamine form hydrogen bond with the inorganic skeleton structure, which effectively improves the stability of the compound 2. For compound 3 and 4 without the organic template, the benzylamine and 2,2′-bipyridine mainly serve as structure-directing agents. Although organic templates do not enter the structure, they are indispensable, because target product cannot be synthesized without organic templates. The two organic amines are selective to the synthesized structure.

## 4. Conclusions

Four different dimensional metal phosphites/phosphates have been synthesized by the mixed solvothermal method in the presence of organic amines. These amines show different mechanisms in the four compounds. The 2,2′-bipyridine organic template acts as a template source and breaks down small molecules in compound 1. For compound 2, 1,2-propanediamine has a role to play as a protonated template. The organic amines of compound 3 and 4 play as structure-directing agent. In terms of skeleton structure, compound 3 contains an odd 7-ring channel and In_2_O_10_ cluster, while compound 4 has 3D intersecting channels and left-handed helical chains. The successful synthesis of four compounds is helpful for studying the role of organic amines and exploring more microporous materials with novel structures.

## Figures and Tables

**Figure 1 materials-13-01752-f001:**
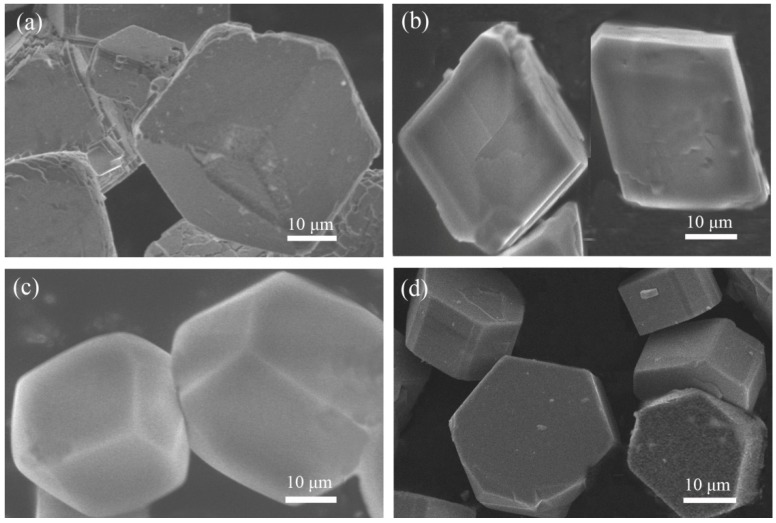
Scanning electron microscopy (SEM) images of compound (**a**) **1**, (**b**) **2**, (**c**) **3**, and (**d**) **4**.

**Figure 2 materials-13-01752-f002:**
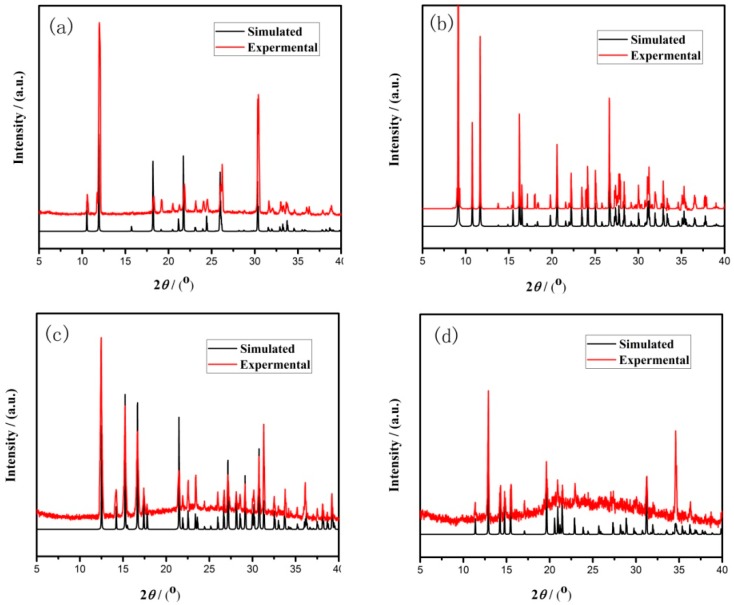
Simulated and experimental X-ray diffraction (XRD) patterns of compound (**a**) **1**, (**b**) **2**, (**c**) **3**, and (**d**) **4**.

**Figure 3 materials-13-01752-f003:**
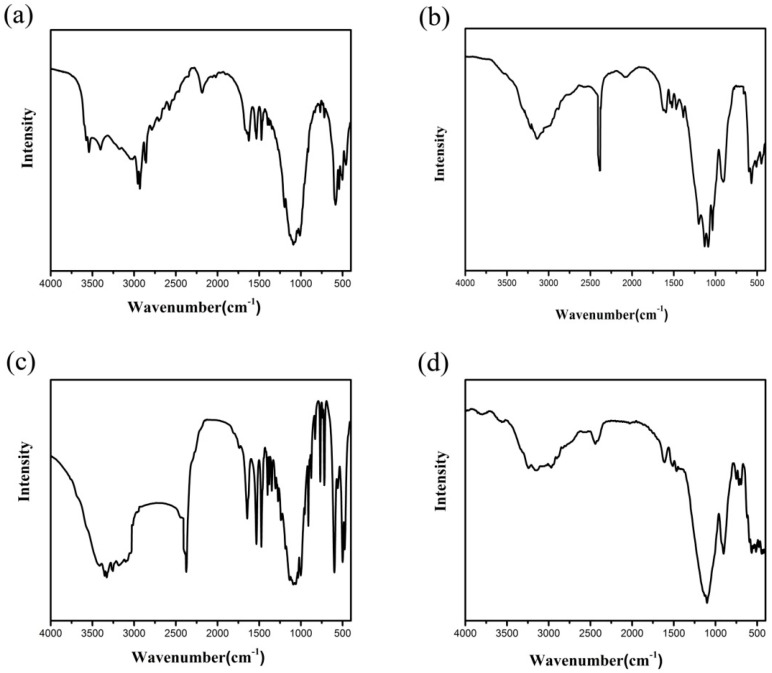
Infrared (IR) spectra of compound (**a**) **1**, (**b**) **2**, (**c**) **3**, and (**d**) **4**.

**Figure 4 materials-13-01752-f004:**
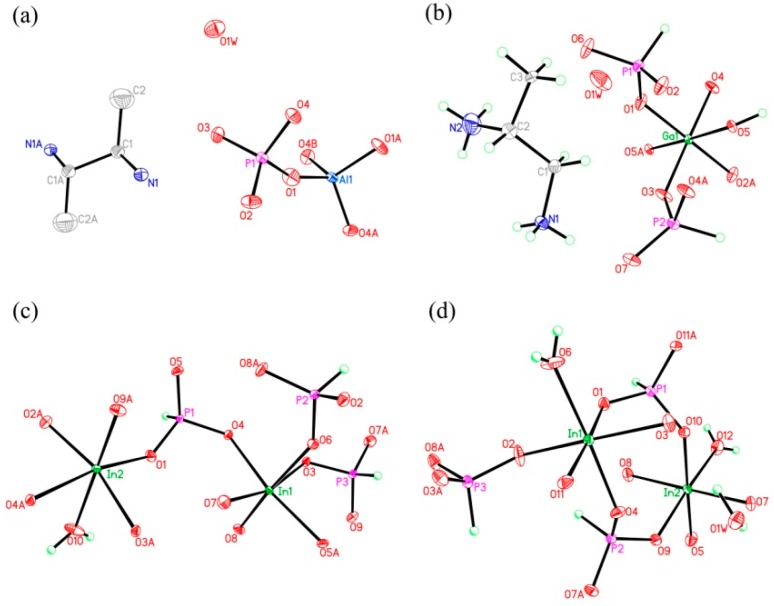
The atomic labelling schemes and thermal ellipsoid plots (30%) of compound (**a**) **1**, (**b**) **2**, (**c**) **3**, and (**d**) **4**.

**Figure 5 materials-13-01752-f005:**
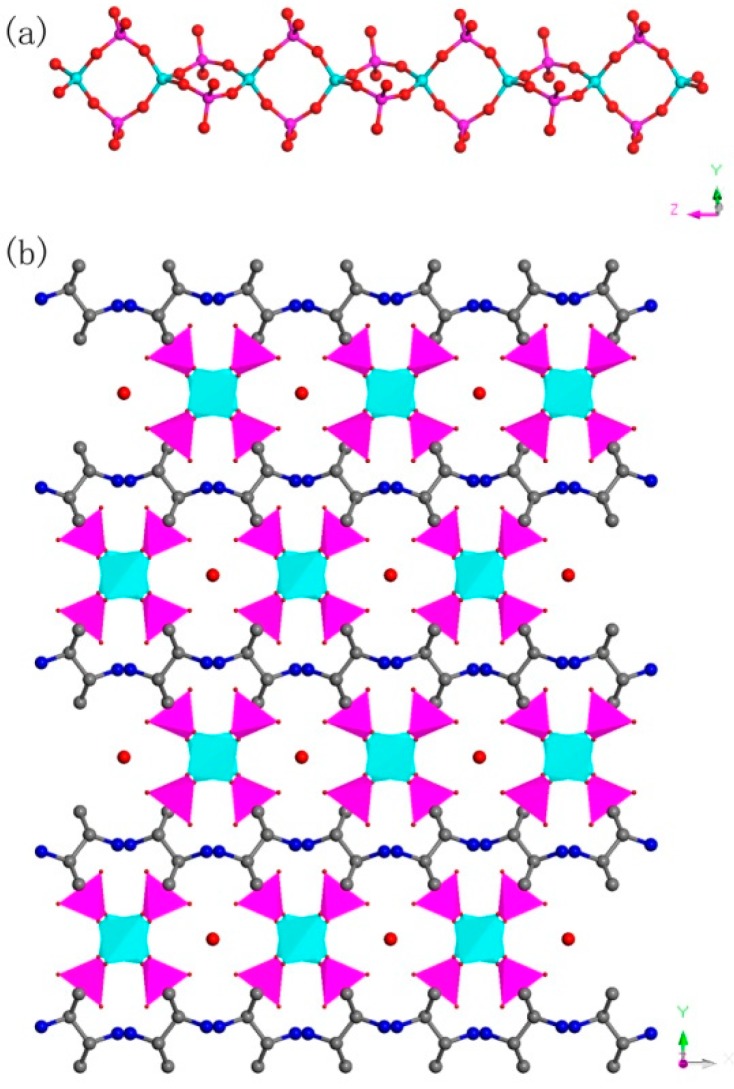
(**a**) One-dimensional (1D) chain structure of **1** along the (001) direction. (**b**) The ABAB dislocated stacking structure of 1 along (010) direction. P, pink; Al, cyan; N, blue; O, red; C, gray. All hydrogen atoms of organic amine are omitted for clarity.

**Figure 6 materials-13-01752-f006:**
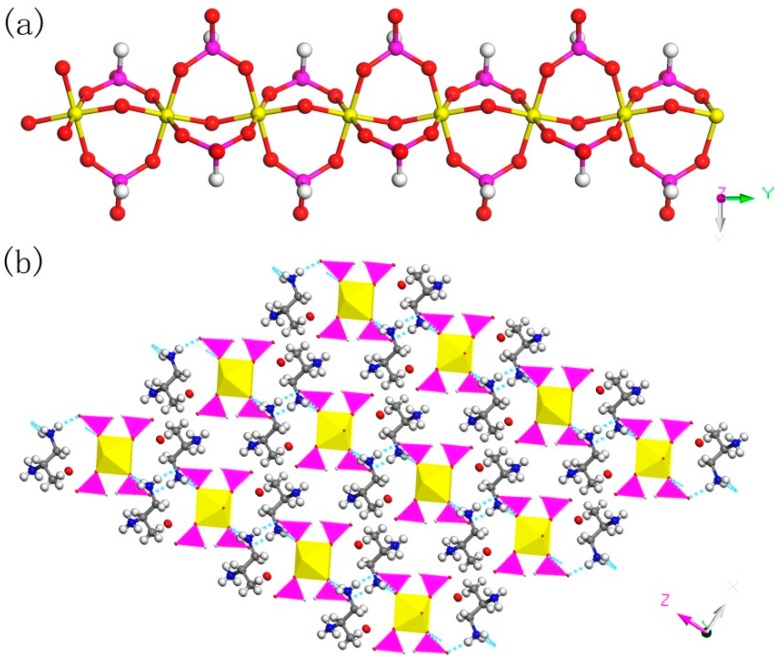
(**a**) 1D chain structure of **2** along the (010) direction. (**b**) The pseudo-2D structure of **2** with hydrogen bond connections. P, pink; Ga, yellow; N, blue; C, gray; H, white; O, red.

**Figure 7 materials-13-01752-f007:**
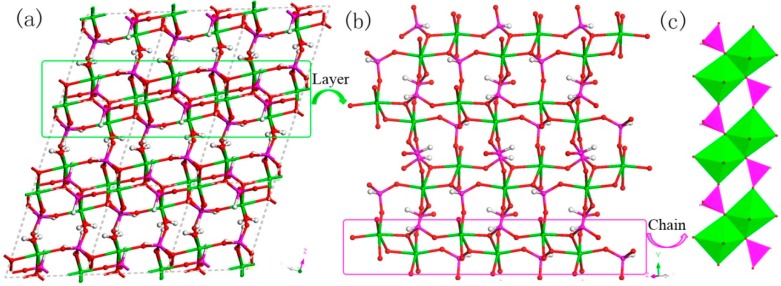
(**a**) The pillared-layer structure of **3** along (001) direction. (**b**) The dense four-ring layer structure of **3** in (101) plane. (**c**) One-dimensional (1D) four-ring chain containing In_2_O_10_ cluster. P, pink; In, green; H, white; O, red.

**Figure 8 materials-13-01752-f008:**
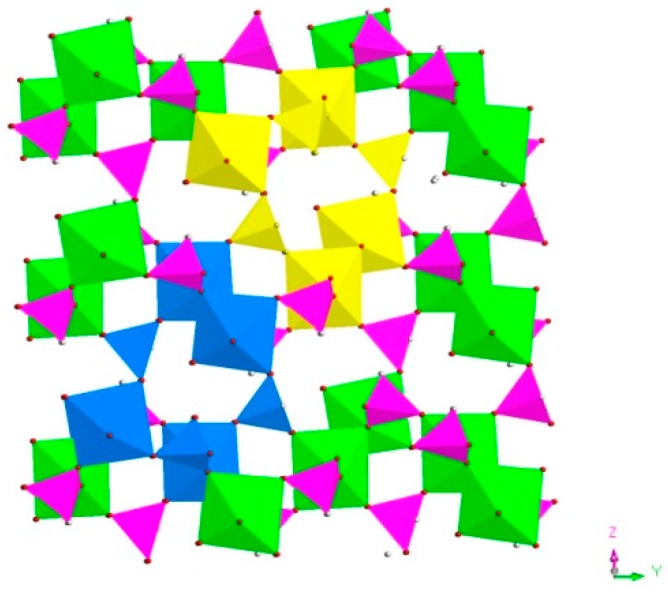
The seven-ring channels in the (100) direction. P, pink; In, green; H, white; O, red; seven-ring channels, yellow and blue.

**Figure 9 materials-13-01752-f009:**
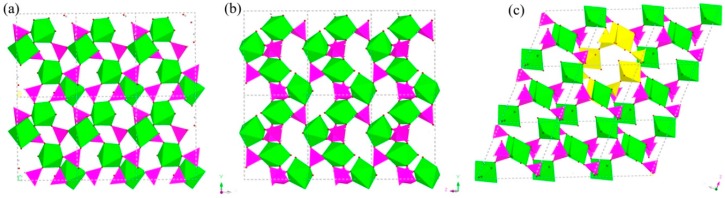
The polyhedron structure of 4 (**a**) six-ring channel in (001) direction. (**b**) Eight-ring channel in (100) direction. (**c**) 10-ring channel in (010) direction. P, pink; In, green; H, white; O, red.

**Figure 10 materials-13-01752-f010:**
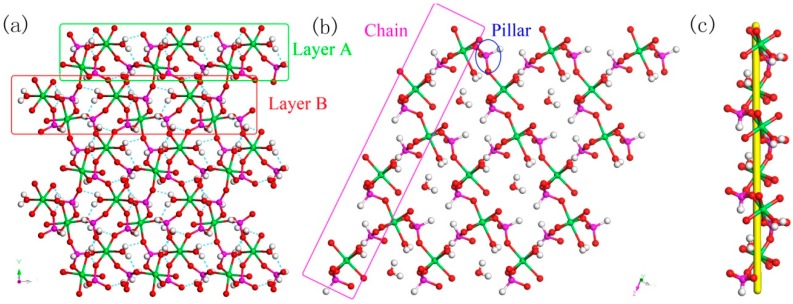
(**a**) The ABAB layer structure of **4** along (001) direction. (**b**) The 12-ring windows of 4 in (101) plane. (**c**) The left-handed helical chain of **4**. P, pink; In, green; H, white; O, red.

**Figure 11 materials-13-01752-f011:**
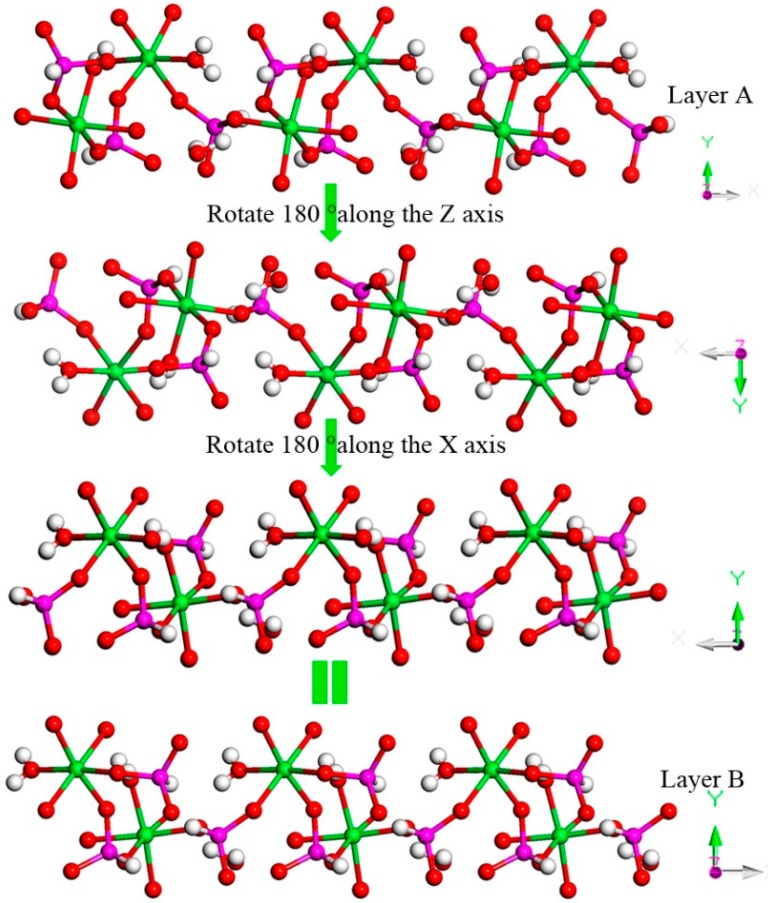
The relationship between AB layers. P, pink; In, green; H, white; O, red.

**Figure 12 materials-13-01752-f012:**
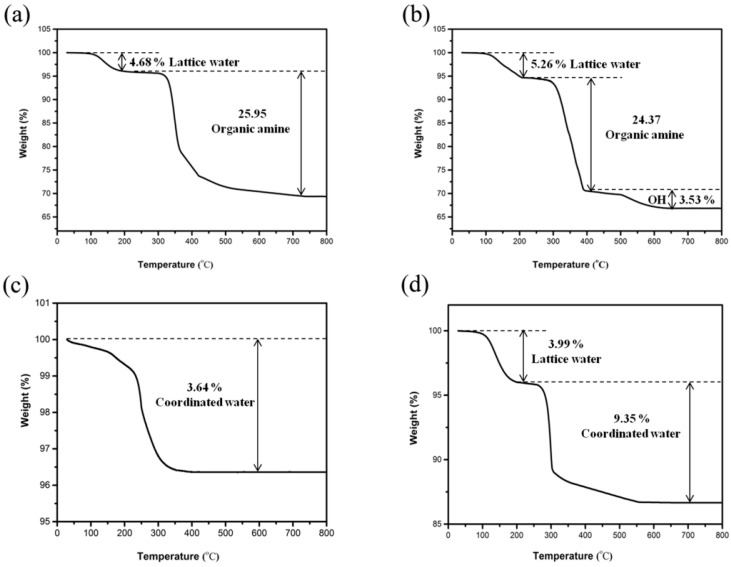
Thermal gravimetric (TG) curves of compound (**a**) 1, (**b**) 2, (**c**) 3, and (**d**) 4.

**Table 1 materials-13-01752-t001:** Crystal data and structure refinement for compounds 1–4.

	1	2	3	4
Empirical formula	C_8_N_4_H_34_Al_2_P_4_O_18_	C_3_N_2_H_17_GaP_2_O_8_	H_5_In_2_P_3_O_10_	H_11_In_2_P_3_O_13_
Formula weight	651.96	340.83	487.59	541.62
Temperature	293(2) K	293(2) K	293(2) K	296(2) K
Wavelength	0.71073 Å	0.71073 Å	0.71073 Å	0.71073 Å
Crystal system, space group	Orthorhombic, *P*ccn	Monoclinic, *P*2_1_/c	Monoclinic, *P*2_1_	Monoclinic, *P*2_1_
Unit cell dimensions	*a* = 8.2701(12)Å *b* =16.791(2) Å *c* = 8.6711(13) Å *α* = 90.00° *β* = 99.110(4)° *γ* = 90.00°	*a* = 11.084(3) Å *b* = 7.093(2) Å *c* = 17.352(4) Å *α* = 90.00° *β* = 119.127(14)° *γ* = 90.00°	*a* = 6.6057(5) Å *b* = 10.1900(8) Å *c* = 7.5004(6) Å *α* = 90.00° *β* = 109.4560(10)° *γ* = 90.00°	*a* = 8.0064(5) Å *b* = 10.3793(6) Å *c* = 8.4640(5) Å *α* = 90.00° *β* = 113.3540(10)° *γ* = 90.00°
Volume	1204.1(3) Å^3^	1191.7(5) Å^3^	476.04(6) Å^3^	645.74(7) Å^3^
*Z*, Calculated density	2, 1.704 Mg/m^3^	4, 1.889 Mg/m^3^	2, 3.402 Mg/m^3^	2, 2.775 Mg/m^3^
Absorption coefficient	0.474 mm^−1^	2.606 mm^−1^	5.385 mm^−1^	4.001 mm^−1^
*F*(000)	612	688	456	512
Crystal size	0.21 × 0.20 × 0.18 mm^3^	0.22 × 0.20 × 0.19 mm^3^	0.21 × 0.20 × 0.18 mm^3^	0.21 x 0.20 x 0.18 mm^3^
Theta range for data collection	2.43 to 28.38°	2.10 to 28.42°	2.88 to 28.31°	2.62 to 28.33°
Limiting indices	−11 ≤ *h* ≤ 10, −22 ≤ *k* ≤ 22, −11 ≤ *l* ≤ 9	−14 ≤ *h* ≤ 14, −7 ≤ *k* ≤ 9, −14 ≤ *l* ≤ 23	−7 ≤ *h* ≤ 8, −12 ≤ *k* ≤ 13, −10 ≤ *l* ≤ 7	−10 ≤ *h* ≤ 10, −10 ≤ *k* ≤ 13, −10 ≤ *l* ≤ 11
Reflections collected/unique	8066/1502 [*R*(int) = 0.0390]	8121/2957 [*R*(int) = 0.0423]	3502/2023 [*R*(int) = 0.0182]	4778/2575 [*R*(int) = 0.0235]
Completeness to theta = 28.31	99.8%	98.6%	99.9%	100.0%
Refinement method	Full-matrix least-squares on *F*^2^	Full-matrix least-squares on *F*^2^	Full-matrix least-squares on *F*^2^	Full-matrix least-squares on *F*^2^
Final *R* indices [*I* > 2σ(*I*)]	*R*_1_ = 0.0591, *wR*_2_ = 0.1929	*R*_1_ = 0.0882, *wR*_2_ = 0.2064	*R*_1_ = 0.0162, *wR*_2_ = 0.0408	*R*_1_ = 0.0209, *wR*_2_ = 0.0470
*R* indices (all data)	*R*_1_ = 0.0698, *wR*_2_ = 0.2029	*R*_1_ = 0.0945, *wR*_2_ = 0.2075	*R*_1_ = 0.0163, *wR*_2_ = 0.0409	*R*_1_ = 0.0219, *wR*_2_ = 0.0474

*R*_1_ = ∑|| *F*_o_ | − | *F*_c_ ||/∑| *F*_o_ |. *wR*_2_ = {∑[*w*(*F*_o_^2^ − *F*_c_^2^)^2^]/∑[*w*(*F*_o_^2^)^2^]}^1/2^.

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
