# Peer review of "The Different Effects of Organic Amines on Synthetic Metal Phosphites/Phosphates"

_materials, 2020, doi:10.3390/ma13071752_

Round 1
Reviewer 1 Report
The Authors proposed a very interesting topic to be presented in the article. The article was prepared carefully and was organized well. The abstract contains relevant information about the scope of article. The paper is publishable, but a minor revision is required before the manuscript acceptance:
- Please, as the last paragraph in the theoretical part, write what was the main purpose of the research, what distinguishes it from those that have already been done.
- Line 52 - 54 - literature is required.
- Line 62, 157, 164, 165, 179 - text editing required.
- Line 61 -66 - please write where the used reagents came from, where they were purchased including the location (manufacturer).
- Line 67 – 75 please write the type and manufacturer of devices used their location (manufacturer).
- What was the experiment measurement error? In how many repetitions was it performed?
- Discussion of results with other similar studies is required.
Reviewer 2 Report
The different effects of organic amines on synthetic metal phosphites/phosphates
Xuelei Wang, Zhaojun Dong, Chao Meng, Wei Wang, Hairui Yang, Xizhun, Zhuo and Shaobin Yang
In this review, authors want to overview the different effects of organic amines on synthetic metal phosphites/phosphates.
The submitted manuscript is interesting, original and in the scoop of the journal.
- The main question: why did the authors use different amines in all four cases? I believe that the role of the amine can be evaluated solely on the example of one compound. Here, the authors used 4 compounds and 4 different amines. This assessment of the role of the amine in my opinion is clearly ambiguous.
- Line 105-107: The TG curve of result showed a total weight loss of 33.17% between 100 and 650 ℃ (Fig. 3b), which was related to the lattice water molecules (calc. 5.29 %), 1,2-propanediamine molecules (calc. 21.75 %) and hydroxyl groups (calc. 4.99 %). Were MS spectra made to confirm the removal of precisely these molecules? Why do the authors believe that it is water and propanediamine are removed?
- What’s more, the authors give some potential applications about the it, why not give a simple example.
- Links are presented in a different format. Use either the full name of the journal or the abbreviation (2,7,13,14 etc.)
I consider this research article of Xuelei Wang, Zhaojun Dong, Chao Meng, Wei Wang, Hairui Yang, Xizhun, Zhuo and Shaobin Yang on novelty and relevance can be recommended for the publication in the Materials (MDPI) after major revision.
Reviewer 3 Report
The article describes four novel metal phosphites and phosphates solvothermally synthesized by organic amines. The material can be interested for chemists. The major revision is necessary nevertheless.
I'm not a specialist in English, but the text contains numerous mistakes and should be thoroughly edited and corrected.
In the abstract: the sentence "The four compounds are analyzed by XRD, SEM, CHN, ICP, IR and TG characterizations" - makes no sense - remove last word "characterizations"
Line 61 - where the substrates were purchased? The supplier should be provided for each reagent.
Line 81 - "the product is pure" or "the product has a pure phase" - decide
Line 84 - Termogravimetric analysis is to shallow. First of all, the derivative of losses should be presented at plots. Each loss region should be considered and clarified. For next samples the same.
Rest of the analysis is done correctly.
Round 2
Reviewer 1 Report
I recommend for publication.
Reviewer 2 Report
Accept in present form
Reviewer 3 Report
All my remarks have been addressed correctly. The manuscript looks much better in the present form, therefore I recommend to publish the article in the present form.